# Relationship between low-back pain and flexibility in children: A cross-sectional study

Tadashi Ito[1,2]*, Hideshi Sugiura[2], Yuji Ito[3,4], Sho Narahara[4], Kentaro Natsume[2], Daiki Takahashi[2], Koji Noritake[5], Kazunori Yamazaki[6], Yoshihito Sakai[7], Nobuhiko Ochi[4]

1 Three-Dimensional Motion Analysis Laboratory, Aichi Prefectural Mikawa Aoitori Medical and Rehabilitation Center for Developmental Disabilities, Okazaki, Japan, 2 Department of Integrated Health Sciences, Graduate School of Medicine, Nagoya University, Nagoya, Japan, 3 Department of Pediatrics, Nagoya University Graduate School of Medicine, Nagoya, Japan, 4 Department of Pediatrics, Aichi Prefectural Mikawa Aoitori Medical and Rehabilitation Center for Developmental Disabilities, Okazaki, Japan, 5 Department of Orthopedic Surgery, Aichi Prefectural Mikawa Aoitori Medical and Rehabilitation Center for Developmental Disabilities, Okazaki, Japan, 6 Institutional Research Center, Aichi Mizuho College, Nagoya, Japan, 7 Department of Orthopedic Surgery, National Center for Geriatrics and Gerontology, Obu, Japan

* sanjigen@mikawa-aoitori.jp

**Data Availability Statement:** If the data are all contained within the manuscript and/or Supporting Information files, enter the following: All relevant

## Abstract

Low-back pain is common among school-aged children. Decreased trunk flexibility in childhood influences low-back pain in adulthood. Previous studies examining the association between low-back pain and trunk flexibility in children are insufficient. Examining this association among elementary school children may help to better understand trunk flexibility in children with low-back pain and to modify the management of inflexibility. Therefore, this study aimed to identify the prevalence of low-back pain and its relationship with physical function among elementary school students. School-aged children aged 6–12 years were recruited in Japan between May 2018 and March 2023. Fingertip-to-floor distance, back muscle strength, pelvic tilt angle during gait, and the visual analog scale for low-back pain were measured. In addition, factors independently related to low-back pain were determined through logistic regression analysis. Low-back pain was reported in 9.6% of the 394 participants (boys, 191; girls, 203). All children with low-back pain presented with back pain when they moved; however, the pain was non-specific. Logistic regression analysis showed that the fingertip-to-floor distance was an independent risk factor for low-back pain (odds ratio, 0.921; p = 0.007). The odds ratios calculated in the logistic regression analysis confirmed that low-back pain frequency increased as the fingertip-to-floor distance decreased. The risk of low-back pain was associated with inflexibility, regardless of sex and muscle strength. These findings suggest that children with low-back pain must increase their trunk and lower extremity flexibility.

## Introduction

Low-back pain is common in both children and adults [1]. Furthermore, back pain in school-aged children is related to the corresponding figure in adulthood [2]. Back pain prevalence

data are within the manuscript and its Supporting Information files.

**Funding:** The authors received no specific funding for this work.

**Competing interests:** The authors have declared that no competing interests exist.

increases with age, from 1% at age 7 to 6% at age 10 [3]. In a large survey of elementary and junior high school students in Japan, back pain was observed in 10.2% [4]. Therefore, low-back pain in this age group is likely a significant risk factor for developing low-back pain in adulthood [5, 6]. The risk factors and prevention strategies for low-back pain in children have not been studied in detail [7–9]. Moreover, the prevalence of back pain is considered to be increasing, especially between 11 and 12 years of age [10, 11]. A recent three-year study of spinal pain in 9-year-old elementary school children reported that both adolescent development and linear growth were associated with spinal pain [12]. Low-back pain in childhood is influenced by body mass index (BMI), physical activity, screen time, and sleep time; the clinical features are diverse [13, 14].

Recent studies have reported that trunk inflexibility affects low-back pain in adulthood [15, 16]. A relationship between low-back pain and hip flexibility has also been reported [17]. Lloyd et al. reported that children's musculoskeletal structures are not adequately developed to support rapid changes in mechanical loading of the spine caused by the differential growth rates of the legs and trunk [18]. Hamstring inflexibility has been associated with low-back pain in adolescence and adulthood [19–21]; however, information is still lacking in studies of elementary school-aged children. Studies on muscle strength have demonstrated no association with low-back pain in healthy children 10–16 years of age [22]. Moreover, tight hamstring and quadriceps muscles have been reported to cause a posterior pelvic tilt in children and adolescents [23]. Children with low-back pain have shown significant improvements in hamstring flexibility with exercise and physical activity [24]. Other systematic reviews have demonstrated that there is a lack of understanding of the relationship between motor function levels, such as flexibility and muscle strength, and low-back pain, as well as the specifics by sex and age group [25]. Investigating the relationship between low-back pain and trunk flexibility in children will help provide new information about factors contributing to low-back pain in children. Based on these previous studies, it is hypothesized that children with back pain may be less flexible due to pain than those without back pain. Thus, understanding the relationship between the flexibility of trunk function and low-back pain may be helpful in the treatment of low-back pain in children. Furthermore, clarifying the association between the risk of low-back pain and flexibility may lead to the development of preventative strategies.

These findings highlight the importance of assessing spinal flexibility in children. Nevertheless, previous studies examining the association between low-back pain and trunk flexibility in children were insufficient. Recent reviews have concluded that the physical performance of the trunk and risk factors for low-back pain in children remain uncertain [10, 13]. Few studies have investigated the differences in flexibility among children in Japan with and without low-back pain by applying flexibility assessments. Additionally, no studies in Japan include flexibility, back strength, or pelvic tilt. Therefore, examining this association in elementary school children would help to better understand trunk flexibility in children with low-back pain and modify the management of inflexibility.

This study aimed to explore the association between low-back pain and flexibility in children and compare these parameters with those in the same age group without low-back pain.

## Materials and methods

### Study population

Three of the 48 Okazaki municipal elementary schools were referred by the Okazaki Board of Education, and they provided informed consent to participate in this study. In total, 585 school-aged children (6–12 years) were recruited for this cross-sectional study between May 2018 and March 2023. Exclusion criteria included orthopedic (n = 49), neurological (n = 2),

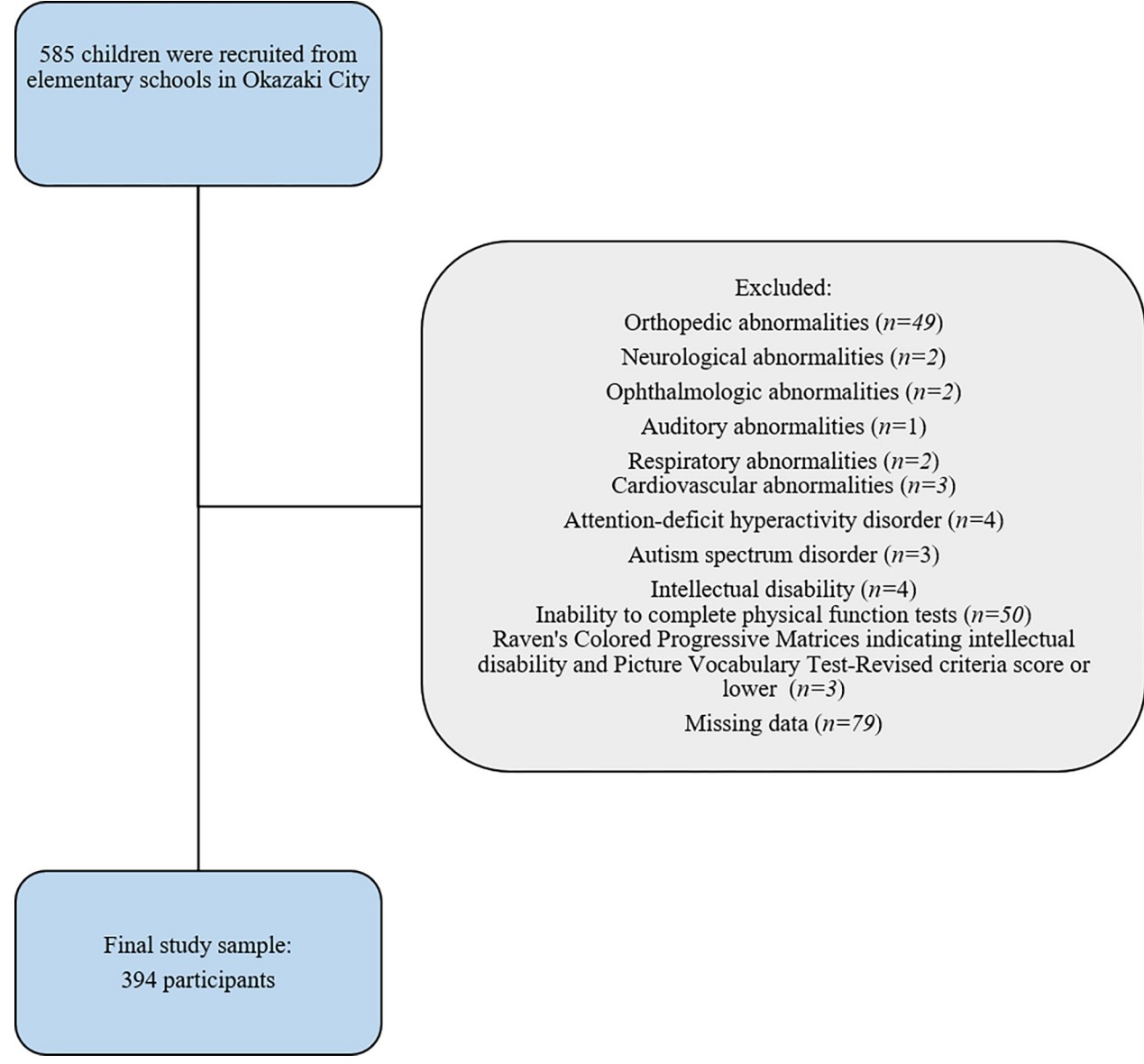

**Fig 1. Flowchart of the enrollment of study participants.**

ophthalmologic (n = 2), auditory (n = 1), respiratory (n = 2), or cardiovascular abnormalities (n = 3) that could affect physical function test results; inability to complete physical function tests (n = 50); Raven's Colored Progressive Matrices indicating intellectual disability and Picture Vocabulary Test-Revised [26, 27] criteria score or lower (n = 3); unable to perform a physical function assessment due to low-back pain (n = 0); and missing data (n = 79) (**Fig 1**). Of the 585 candidates, 191 were excluded, and 394 were enrolled in the study.

Low-back pain was assessed using a visual analog scale (0–10 cm). An orthopedic surgeon interviewed the participants to determine whether low-back pain occurred during movement [28]. Based on the pain severity evaluation by Boonstra et al., low-back pain severity was defined as ≥3 cm of symptoms in low-back pain [29]. Higher scores indicate more severe low-back pain. Based on this score, the participants were divided into children with low-back pain (score ≥3 cm) and those without low-back pain (score 0–2.9 cm).

This study was conducted in accordance with the Declaration of Helsinki, and the protocol was approved by the Ethics Committee of the Aichi Prefectural Mikawa Aoitori Ethics Review Board (approval number: 29002). The authors did not have access to information that could identify individual participants during or after data collection. The legal guardians of all participants provided written informed consent for their children to participate in the study and to release identifying information. All children consented to participate in the study.

## Data collection

**Questionnaire.** Moderate physical activity hours per week were assessed based on the Physical Activity Index recommended by the World Health Organization [30]. Children provided self-reported responses to the physical activity time questionnaire. Screen time was investigated by asking children to fill in the daily hours spent watching TV and movies. Their responses were verified by their parents. Parents and children rated sleep hours per day in relation to sleep history [30]. Furthermore, an orthopedic surgeon interviewed the participants to determine whether low-back pain occurred during movement. Low-back pain was assessed using a visual analog scale (0–10 cm). The participants were asked to perform trunk flexion and extension in a standing position to confirm low-back pain during movement [28]. In addition, for low-back pain, the patient was asked about the current pain level and when it began to manifest [28].

**Measurement of the finger–floor distance.** The finger–floor distance (cm), a measure of flexibility, was evaluated using a digital trunk forward flexion meter (FLEXION-D; Takei Ltd., Niigata, Japan). The finger–floor distance was measured from the tips of the middle fingers to the floor in barefoot children with their knees straight and feet placed together and bent maximally forward (**Fig 2**) [31, 32]. When the tip of the middle finger reached the floor, 0 cm was indicated, and the absolute value increased with increasing distance from 0 cm. Negative values were obtained when the tip did not reach the floor. Measurements were recorded twice and in 0.1 cm increments, and the best result out of the two was considered the measured value. The shorter the finger–floor distance (fingertip not reaching the floor; negative value), the lower the flexibility. In a previous study, inter-rater reliability analysis for forward flexion measured by finger–floor distance had an acceptable intraclass correlation coefficient of 0.99 [33].

**Back muscle strength.** A digital back dynamometer (Back-D; Takei Ltd., Niigata, Japan) was used to measure back muscle strength [31, 34]. Participants were asked to stand on a platform with their feet shoulder-width apart. For adjustment by the research assistant, the chain length was adjusted according to the height difference. Back muscle strength was determined based on the maximum isometric extension force of the trunk muscles in the standing position, with the trunk flexed at 30˚ [31, 34]. This was undertaken to ensure that the force came from the back muscles, not the psoas or shoulder muscles, so the participant did not fall backward. Measurements were performed twice, and the average of the two values was recorded. Data on back muscle strength were corrected for weight. The reliability of the measurement tools evaluated by the intraclass correlation coefficient ranged from 0.93 to 0.97 [35].

## Pelvic tilt during gait

A physical therapist with more than 10 years of experience in clinical gait analysis performed the three-dimensional gait analysis using a motion analysis system (MX-T 20S; Vicon, Oxford, UK) with eight cameras and a sampling frequency of 100 Hz. The plug-in-gait model (May 2018 to March 2020) and Conventional Gait Model 2.3 (June 2020 to March 2023), with markers placed on the lower body, were used to measure gait [30, 36]. The children walked barefoot

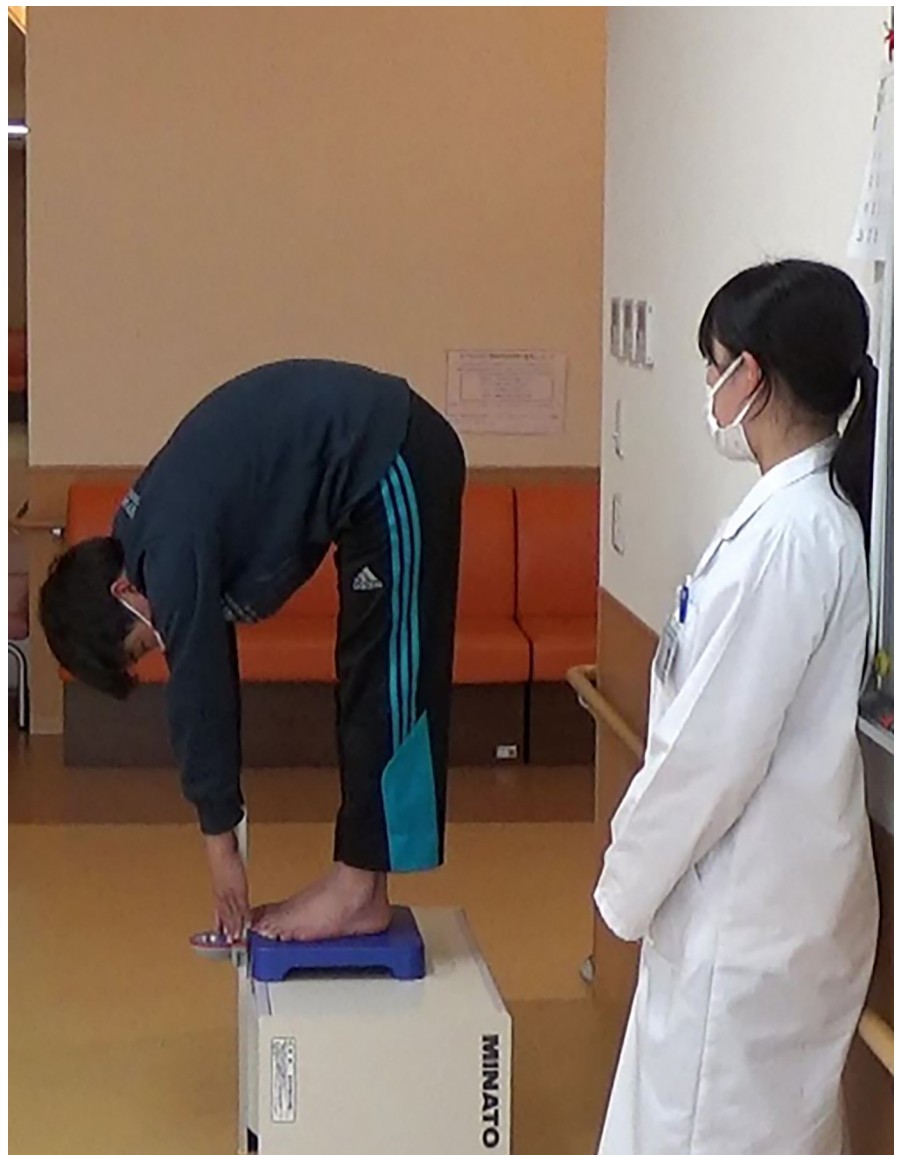

**Fig 2. Measuring the fingertip-to-floor distance using a digital trunk forward flexion meter.** If the fingertip did not reach the floor, the fingertip-to-floor distance was assigned a negative value; if it exceeded the distance to the floor, the distance was assigned a positive value.

at a self-regulated speed on an 8-AMTI OPT force plate (Advanced Mechanical Technology, Inc., Watertown, MA, USA) in three trials [36]. Pelvic tilt was assessed using the maximum value in the sagittal plane during the stance phase. The mean maximum value of the pelvic tilt was calculated using the results of three gait trials analyzing the right and left lower legs [36].

## Sample size

The sample size was determined using G*Power (Heinrich Heine University, Düsseldorf, Germany) [37, 38]. Based on previous research, we considered the proportion of children with low-back pain to be 0.1 [4], with a statistical power of 0.8, two-tailed alphas of 0.05, and a medium effect size (d = 0.5). Based on these assumptions, the required sample size was 382 (35 children with low-back pain and a control group of 347).

## Statistical analysis

The normality of the distribution of each variable was checked using the Shapiro-Wilk test. The chi-squared test was used to compare the differences in the proportions of each sex in each group. Participant data are expressed as a mean (standard deviation) or median (range) and compared using an independent t-test or Mann–Whitney U test, as appropriate. Logistic regression was used to assess the association between finger–floor distance, back muscle strength, pelvic tilt during gait, and low-back pain. Statistical significance was defined as a two-sided p-value of <0.05. Multivariate logistic regression was used to determine the odds ratios of trunk function associated with low-back pain after controlling for sex as a confounding factor. The group (children with or without low-back pain) was the dependent variable in this analysis. All data analyses were performed using SPSS version 28.0 (IBM Corp., Armonk, NY, USA).

## Results

This study included 394 participants (191 boys and 203 girls; 38 with low-back pain and 356 without low-back pain) with a mean age of 8.9 (range, 6–12) years who were admitted for medical examination and physical function evaluation. All children with low-back pain had non-specific low-back pain, and 97.3% reported low-back pain during movement. Furthermore, none of the participants answered clearly regarding when the low-back pain began. **Tables 1 and 2** show the study participants' demographic characteristics, trunk function measures, and a comparison between the two groups. Children with low-back pain were found to have reported a larger visual analog scale score (p < 0.0001; 95% confidence interval [CI]: -4.000 to -3.500) than those without low-back pain (**Table 1**). Furthermore, there was a significant difference based on sex (p = 0.025; 95% CI: 1.091–4.440), with more boys reporting low-back pain (65.8%). There were no significant differences based on age, height, weight, BMI, physical activity time, screen time, or sleep time between the groups with and without low-back pain. Children with low-back pain had shorter finger–floor distances (p = 0.003; 95% CI: 1.183–5.586) than those without low-back pain (**Table 2**). The two groups had no significant differences in back muscle strength or pelvic tilt during gait.

Multivariate logistic regression analyses for low-back pain, performed among trunk functions, showed that the finger–floor distance (odds ratio 0.921; P = 0.007; 95% CI: 0.868–0.977)

**Table 1. Demographic characteristics of participants with and without low-back pain.**

| Variable | Children with low-back pain (n = 38) | Children without low-back pain (n = 356) | p-value | 95% CI |
|---|---|---|---|---|
| Age (years), median (range) | 9 (7–12) | 9 (6–12) | 0.182 | -1.000–0.000 |
| Sex, n (%) | | | 0.025 | 1.091–4.440 |
| Female | 13 (34.2) | 190 (53.4) | | |
| Male | 25 (65.8) | 166 (46.6) | | |
| Height (cm), median (range) | 132.7 (110.7–158.7) | 130.4 (107.5–173.2) | 0.306 | -6.400–1.900 |
| Weight (kg), median (range) | 27.6 (18.1–45.7) | 27.2 (16.2–77.9) | 0.407 | -3.300–1.400 |
| Body mass index (kg/m$^2$), median (range) | 15.9 (13.4–21.7) | 15.8 (12.2–30.2) | 0.940 | -0.580–0.547 |
| Physical activity (hours), median (range) | 4.8 (0–21) | 4.0 (0–22) | 0.715 | -1.000–1.500 |
| Screen time per day (hours), median (range) | 2.0 (0–5) | 1.6 (0–9) | 0.219 | -0.500–0.000 |
| Sleep time per day (hours), median (range) | 9 (6–10) | 9 (6.5–11) | 0.415 | 0.000–0.000 |
| Visual analog scale (cm), median (range) | 4 (3–10) | 0 (0–2) | 0.0001 | -4.000 –-3.500 |

The p-value for the difference in the proportion by sex was calculated using the chi-squared test, and the other p-values were calculated using the Mann–Whitney U test.

CI, confidence interval

**Table 2. Physical performance of the trunk in participants with and without low-back pain.**

| Variable | Children with low-back pain (n = 38) | Children without low-back pain (n = 356) | p-value | 95% CI |
|---|---|---|---|---|
| Fingertip-to-floor distance (cm) | 1.9 (7.2) | 5.2 (6.5) | 0.003 | 1.183–5.586 |
| Back muscle strength (kg/kg) | 1.2 (0.6–2.0) | 1.1 (0.6–2.2) | 0.287 | -4.750–1.500 |
| Pelvic tilt during gait (degree) | 15.5 (4.3) | 15.6 (4.3) | 0.859 | -1.307–1.568 |

Data are presented as a mean (standard deviation) or median (range).

The p-values of the fingertip-to-floor distance and pelvic tilt were calculated using the independent t-test, and the other p-values were calculated using the Mann–Whitney U test.

CI, confidence interval

was associated with low-back pain. In contrast, back muscle strength and pelvic tilt during walking were not associated with low-back pain (**Table 3**).

## Discussion

The current study aimed to examine the association between flexibility and low-back pain in children. The key finding was that flexibility was associated with low-back pain. In this cross-sectional study, using the odds ratio of a multivariable logistic regression, we observed that each additional shorter finger–floor distance was related to a higher frequency of reporting low-back pain.

Previous studies reported decreased flexibility of the hamstring and quadriceps muscles in adolescents with low-back pain [21, 39]. Furthermore, stiffness of the thoracolumbar fascia results from rapid growth and the development of low-back pain due to preexisting inflexibility [40]. However, another study reported that flexibility and muscle strength were not associated with low-back pain [41]. The studies analyzing the relationship between low-back pain and flexibility in children are lacking, and further investigation, including longitudinal studies, is warranted. The finger–floor distance was shorter in children with low-back pain than in those without low-back pain. The results of the current study showed no differences in age between groups, suggesting that low flexibility is associated with low-back pain in 6–12-year-old children. Thus, poor flexibility of the trunk and lower extremities may be related to low-back pain in the pediatric population. Moreover, the differences in these results may be due to differences in race and lifestyle.

Several reports have demonstrated that low-back pain in children is as frequent a complaint as in adults [3, 4, 42]. In another study, the incidence of low-back pain in children was 10.2% [4], which was similar to the 9.6% result in our study. Furthermore, low-back pain during movement accounted for 97.3% of all non-specific low-back pain, indicating that the cause of

**Table 3. Relationship between the physical performance of the trunk and low-back pain.**

| Variable | β | SE | Wald | Odds ratio (95% CI) | p-value |
|---|---|---|---|---|---|
| Fingertip-to-floor distance | -0.082 | 0.030 | 7.345 | 0.921 (0.868–0.977) | 0.007 |
| Back muscle strength | 0.919 | 0.641 | 2.052 | 2.506 (0.713–8.810) | 0.152 |
| Pelvic tilt during gait | 0.034 | 0.043 | 0.620 | 1.035 (0.951–1.126) | 0.431 |
| Sex | 0.442 | 0.395 | 1.247 | 1.555 (0.717–3.375) | 0.264 |

Thirty-eight children with and 356 without low-back pain were analyzed. The occurrence of low-back pain was the dependent variable (without low-back pain = 0, with low-back pain = 1), and sex was an adjustment variable.

β, partial regression coefficient; CI, confidence interval; SE, standard error.

the apparent low-back pain is unknown and that evaluation of back pain in children must be done with caution. This term is used when the pathoanatomical cause of pain cannot be identified [43]. Although detailed reports on low-back pain in Japanese elementary school children are scarce, and no definite conclusions may be drawn, our results suggest that low-back pain in Japanese children is more common in boys than in girls. Hence, boys may be exposed to potentially more dangerous and strenuous sporting activities than girls. However, logistic regression analysis did not show an association between sex and low-back pain. Nevertheless, a Spanish study has reported a higher rate of back pain in girls than in boys [9]. Meanwhile, there are reports of an unclear relationship between low-back pain and sex [44]. The differences in these results may be related to lifestyle and exercise habits in different countries [45, 46]. There may also be differences based on the target age group, though these differences have not yet been clarified and require further investigation.

Back muscle strength was not a risk factor for low-back pain in this study, suggesting that weak back muscle strength is not necessarily related to low-back pain. A recent study reported that trunk muscle strength was not associated with low-back pain in children, although extensor muscle endurance was associated with low-back pain [41]. Noll et al. reported that trunk extensor muscle endurance deficits might increase vulnerability to tissue strain during sporting activities [41]. However, present study evaluated back muscle strength rather than endurance, which is a topic for future research.

This study found no association between low-back pain and pelvic tilt during walking. This result was consistent with that of other studies showing a lack of a relationship between pelvic tilt and low-back pain in children [47, 48].

BMI, physical activity, screen time, and sleep time were not associated with low-back pain in the present study's participants. These results were inconsistent with those of previous studies [13, 14]. Low-back pain did not affect these factors in this study, indicating that children with low-back pain do not necessarily exhibit greater changes in BMI, physical activity, screen time, and sleep time. Moreover, it is necessary to consider that the participants' age range and inclusion and exclusion criteria can influence the results.

This study had a few limitations. First, this cross-sectional study only demonstrated the association between flexibility or physical performance of the trunk and low-back pain; therefore, it is impossible to prove an apparent causal association. Second, the low-back pain assessments were based on self-reported data. Third, the study sample was relatively small and included only community-dwelling school-aged children from Japan. However, the results of this study could be used in elementary schools in the community to promote education on low-back pain prevention.

Despite these limitations, the present study had several strengths. This was a cross-sectional study to test the hypothesis that low-back pain in childhood is associated with decreased flexibility. Furthermore, this was the first study in Japan conducted among elementary school children to demonstrate an association between low-back pain and flexibility in childhood, which has not been clearly understood as a distinct characteristic compared with adults and older adults. In addition, study participants were assessed for flexibility using a physical function assessment that could be adopted and reproduced in a clinical setting.

In the present study, the prevalence of low-back pain in children was 9.6%, of which 97.3% had low-back pain during movement, and boys were more likely to have low-back pain. Low-back pain was only associated with flexibility. Back muscle strength, pelvic tilt during gait, and sex were not associated with low-back pain. Therefore, it is important to consider that children who experience low-back pain during exercise are more likely to have reduced flexibility. These findings have implications for preventing low-back pain in children and may provide useful information for clinicians and researchers. However, given our findings, future studies

with larger sample sizes are required to further investigate this issue. If our findings are confirmed in future studies, clinicians, patients, and researchers may consider flexibility in the assessment of low-back pain. At this time, it may be clinically relevant for clinicians and researchers to inform patients that decreased flexibility is a contributing factor to low-back pain. This is because there are rehabilitation options that can help address the symptoms that lead to loss of flexibility.

## Conclusions

The present study demonstrated that the risk for low-back pain was associated with flexibility, regardless of sex and muscle strength, with a shorter finger–floor distance found in children with low-back pain. It indicated that low-back pain is common in boys (65.8%), with an entirety of 9.6%. Improved knowledge and awareness of the risk factors for low-back pain and the need for good flexibility are areas that require further investigation. Interventions to improve flexibility can reduce the degree of pain experienced by children with low-back pain.

## Supporting information

**S1 File.**
(XLSX)

## Acknowledgments

We thank the Aichi Pediatrics Medical Society, Neishi Primary School, Okazaki City Board of Education, Okazaki City Medical Association, Otogawa Primary School, Umezono Primary School, Aichi Prefectural Mikawa Aoitori Medical and Rehabilitation Center for Developmental Disabilities staff, and Nagoya University staff for helping recruit participants.

## Author Contributions

**Conceptualization:** Tadashi Ito, Yuji Ito, Sho Narahara, Daiki Takahashi.

**Data curation:** Tadashi Ito, Hideshi Sugiura, Yuji Ito, Sho Narahara, Kentaro Natsume, Daiki Takahashi, Kazunori Yamazaki, Nobuhiko Ochi.

**Formal analysis:** Tadashi Ito, Kentaro Natsume, Daiki Takahashi.

**Investigation:** Tadashi Ito.

**Methodology:** Tadashi Ito, Sho Narahara, Kentaro Natsume, Yoshihito Sakai.

**Project administration:** Tadashi Ito.

**Resources:** Tadashi Ito, Daiki Takahashi.

**Software:** Tadashi Ito, Kentaro Natsume, Daiki Takahashi.

**Supervision:** Hideshi Sugiura, Koji Noritake, Yoshihito Sakai, Nobuhiko Ochi.

**Validation:** Tadashi Ito, Kentaro Natsume.

**Visualization:** Tadashi Ito, Sho Narahara, Koji Noritake, Kazunori Yamazaki.

**Writing – original draft:** Tadashi Ito.

**Writing – review & editing:** Hideshi Sugiura, Yuji Ito, Sho Narahara, Kentaro Natsume, Daiki Takahashi, Koji Noritake, Kazunori Yamazaki, Yoshihito Sakai, Nobuhiko Ochi.

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
