## [Decision Letter · Decision Letter 0]

31 Jul 2023

PONE-D-23-20765Relationship Between Low-Back Pain and Physical Function in Children: A Cross-Sectional StudyPLOS ONE

Dear Dr. Ito,

Thank you for submitting your manuscript to PLOS ONE. After careful consideration, we feel that it has merit but does not fully meet PLOS ONE’s publication criteria as it currently stands. Therefore, we invite you to submit a revised version of the manuscript that addresses the points raised during the review process.

We look forward to receiving your revised manuscript.

Kind regards,

Mohamed El-Sayed Abdel-Wanis, Ph.D.

Academic Editor

PLOS ONE

At this time, please address the following queries.

5. We note that Figure 1 in your submission contain copyrighted images. All PLOS content is published under the Creative Commons Attribution License (CC BY 4.0), which means that the manuscript, images, and Supporting Information files will be freely available online, and any third party is permitted to access, download, copy, distribute, and use these materials in any way, even commercially, with proper attribution. For more information, see our copyright guidelines: http://journals.plos.org/plosone/s/licenses-and-copyright.

Reviewers' comments:

Reviewer's Responses to Questions

**Comments to the Author**

1. Is the manuscript technically sound, and do the data support the conclusions?

Reviewer #1: Yes

Reviewer #2: Yes

2. Has the statistical analysis been performed appropriately and rigorously? 

Reviewer #1: Yes

Reviewer #2: Yes

3. Have the authors made all data underlying the findings in their manuscript fully available?

Reviewer #1: Yes

Reviewer #2: Yes

4. Is the manuscript presented in an intelligible fashion and written in standard English?

Reviewer #1: Yes

Reviewer #2: Yes

5. Review Comments to the Author

Reviewer #1: Article: PONE-D-23-20765

Relationship Between Low-Back Pain and Physical Function in Children: A Cross- Sectional Study

GENERAL COMMENTS

Thank you for allowing me to review this manuscript. The manuscript adhere the PLOS ONE Data Policy. The aim of this study was to identify the prevalence of low-back pain and its relationship with physical function among elementary school students. This is an interesting research topic with potential utilization across disciplines and relevant to the journal. In my opinion, the paper would need minor changes. Revisions will be necessary. Improve the organization of your paper using the following guidelines.

INTRODUCTION

Abstract: The title of the study and the objective of the study is not matching (include: flexibility) Please revise it.

What's new in the scientific literature with this manuscript? Include in introduction. The use of systematic reviews is recommended.

-The manuscript must be include a hypothesis. Explain the hypothesis.

METHODS

-"All children with low-back pain presented with back pain  when they moved; however, the pain was non-specific". Who has diagnosed the condition and on what basis or criteria has been diagnosed. Include literature references.

-"Of the 585 candidates, 191 were excluded, and 394 were enrolled in the study". Include a flowchart.

- Measurement of the finger–floor distance and back muscle strength: Include Inter-rater reliability analysis. It is important to support the use of the instrument and clearly and objectively justify the reason for performing this type of analysis.

RESULTS

- The results should be presented with 95%CI (upper limit – lower limit) for all the variables.

DISCUSSION

-Include the strengths of the study.

-ARE THE RESULTS APPLICABLE IN PRACTICE? Add. Include the implications/applications of the study for the field movement sciences.

- Include the clinical significance of this study over clinicians, patients, and researchers after the study hypothesis.

Reviewer #2: The manuscript is interesting, well written and objective. The results add new information to the topic, which can be used by researchers and professionals who deal with health and/or human motor behavior.

Some observations were made in order to improve the clarity of the text.

In the Methods section,

*Regarding data reduction, note that different methods were used for each of the three variables:

Lines 137-138: “Measurements were recorded twice and in 0.1 cm increments, and the best result out of the two was considered the measured value.”

Line 156: “Measurements were performed twice, and the average of the two values was recorded.”

Line 167-169: “The mean maximum value of the pelvic tilt was calculated using the results of three gait trials analyzing the right and left lower legs.”

Please clarify for the reader why each of these different data reductions was used or indicate references that have used the same types of reductions.

*Repetition of information in the text and description of the table.

Note that there is redundancy in explaining the contents of Figure 1 in the text (lines 132-134) and below the Table. Please adjust the text below the Table, making it more accurate and concise.

*The formatting of Table 1 needs revision and adjustments, especially considering the text justification. Please see standards https://journals.plos.org/plosone/s/tables

In the Discussion section,

*Between lines 258-264 there is an excessive use of the term “However” which has made the text confusing. Please adjust this part of the text, seeking to express the central idea in a more objective and reader-friendly way.

*In lines 257-258 (“Hence, boys may be exposed to potentially more dangerous and strenuous sporting activities than girls.” and in lines 262-263 (“The differences in these results may be related to lifestyle and exercise habits in different countries) claims are not supported by the results of the present study. It is strongly suggested that references be added to support such claims.

6. PLOS authors have the option to publish the peer review history of their article (what does this mean?). If published, this will include your full peer review and any attached files.

Reviewer #1: **Yes: **Sílvia Maria Amado João

Reviewer #2: **Yes: **Maria Teresa Cattuzzo

---

## [Author Response · Author response to Decision Letter 0]

11 Sep 2023

Reviewer #1: 

Thank you for providing these insights. We wish to express our sincere appreciation for your insightful comments on our paper. We feel the comments have helped us significantly improve the paper.

Comments

Relationship Between Low-Back Pain and Physical Function in Children: A Cross- Sectional Study

GENERAL COMMENTS

Thank you for allowing me to review this manuscript. The manuscript adhere the PLOS ONE Data Policy. The aim of this study was to identify the prevalence of low-back pain and its relationship with physical function among elementary school students. This is an interesting research topic with potential utilization across disciplines and relevant to the journal. In my opinion, the paper would need minor changes. Revisions will be necessary. Improve the organization of your paper using the following guidelines.

Response: Thank you for your support. We acknowledge that the INTRODUCTION, METHODS, RESULTS, and DISCUSSION could be improved, ensuring alignment with our study. To this end, we have fully revised our text according to the reviewers’ comments.

Comments

INTRODUCTION

Abstract: The title of the study and the objective of the study is not matching (include: flexibility) Please revise it.

What's new in the scientific literature with this manuscript? Include in introduction. The use of systematic reviews is recommended.

-The manuscript must be include a hypothesis. Explain the hypothesis.

Response: We agree with this comment and have fully revised the Introduction section to avoid ‘vagueness’. We have also included specific information regarding the new findings in the scientific literature, the importance of this study, and a hypothesis, as follows: 

Children with low-back pain have shown significant improvements in hamstring flexibility with exercise and physical activity [24]. Other systematic reviews have demonstrated that there is a lack of understanding of the relationship between motor function levels, such as flexibility and muscle strength, and low-back pain, as well as the specifics by sex and age group [25]. Investigating the relationship between low-back pain and trunk flexibility in children will help provide new information about factors contributing to low-back pain in children. Based on these previous studies, it is hypothesized that children with back pain may be less flexible due to pain than those without back pain. 

The title has also been revised, and an abstract has been added. 

Comments

METHODS

-"All children with low-back pain presented with back pain when they moved; however, the pain was non-specific". Who has diagnosed the condition and on what basis or criteria has been diagnosed. Include literature references.

-"Of the 585 candidates, 191 were excluded, and 394 were enrolled in the study". Include a flowchart.

- Measurement of the finger–floor distance and back muscle strength: Include Inter-rater reliability analysis. It is important to support the use of the instrument and clearly and objectively justify the reason for performing this type of analysis.

Response: This was originally detailed in the "Materials and Methods" and "Questionnaire" section. For clarity, we have added the orthopedic surgeon’s diagnosis and cited the literature in the Materials and Methods section. The diagnosis of low-back pain was also made by an orthopedic surgeon using a visual analog scale, and the presence or absence of low-back pain was based on previous studies by Boonstra et al.

We also created a flow chart of the enrollment of study participants (Figure 1) and added references for inter-rater reliability of finger-to-ground distance and back muscle strength. Since an inter-rater reliability analysis was impossible in this study, we provided a rationale by citing previous studies.

Comments

RESULTS

- The results should be presented with 95%CI (upper limit – lower limit) for all the variables.

Response: Thank you for providing these insights. We have added 95% confidence intervals to the RESULTS, as suggested.

Comments

DISCUSSION

-Include the strengths of the study.

-ARE THE RESULTS APPLICABLE IN PRACTICE? Add. Include the implications/applications of the study for the field movement sciences.

- Include the clinical significance of this study over clinicians, patients, and researchers after the study hypothesis.

Response: We agree with this comment and have revised the Discussion section to provide a strength of the study. We have also provided a summary of the clinical significance of this study in the field of movement sciences, as follows:

Despite these limitations, the present study had several strengths. This was a cross-sectional study to test the hypothesis that low-back pain in childhood is associated with decreased flexibility. Furthermore, this was the first study in Japan conducted among elementary school children to demonstrate an association between low-back pain and flexibility in childhood, which has not been clearly understood as a distinct characteristic compared with adults and older adults. In addition, study participants were assessed for flexibility using a physical function assessment that could be adopted and reproduced in a clinical setting.

In the present study, the prevalence of low-back pain in children was 9.6%, of which 97.3% had low-back pain during movement, and boys were more likely to have low-back pain. Low-back pain was only associated with flexibility. Back muscle strength, pelvic tilt during gait, and sex were not associated with low-back pain. Therefore, it is important to consider that children who experience low-back pain during exercise are more likely to have reduced flexibility. These findings have implications for preventing low-back pain in children and may provide useful information for clinicians and researchers. However, given our findings, future studies with larger sample sizes are required to further investigate this issue. If our findings are confirmed in future studies, clinicians, patients, and researchers may consider flexibility in the assessment of low-back pain. At this time, it may be clinically relevant for clinicians and researchers to inform patients that decreased flexibility is a contributing factor to low-back pain. This is because there are rehabilitation options that can help address the symptoms that lead to loss of flexibility.

Reviewer #2: 

Thank you for providing us with insightful comments regarding our manuscript. We express our sincere gratitude for taking the time and effort necessary to revise our text. Your comments and suggestions helped us immensely improve our work.

Comments

Reviewer #2: The manuscript is interesting, well written and objective. The results add new information to the topic, which can be used by researchers and professionals who deal with health and/or human motor behavior.

Some observations were made in order to improve the clarity of the text.

Response: Thank you for your detailed review and support. We agree that the methods could have been more clearly presented. We have fully revised our manuscript to address comments and to improve the overall clarity of our study. We have provided detailed responses to the concerns raised.

Comments

In the Methods section,

*Regarding data reduction, note that different methods were used for each of the three variables:

Lines 137-138: “Measurements were recorded twice and in 0.1 cm increments, and the best result out of the two was considered the measured value.”

Line 156: “Measurements were performed twice, and the average of the two values was recorded.”

Line 167-169: “The mean maximum value of the pelvic tilt was calculated using the results of three gait trials analyzing the right and left lower legs.”

Please clarify for the reader why each of these different data reductions was used or indicate references that have used the same types of reductions.

*Repetition of information in the text and description of the table.

Note that there is redundancy in explaining the contents of Figure 1 in the text (lines 132-134) and below the Table. Please adjust the text below the Table, making it more accurate and concise.

*The formatting of Table 1 needs revision and adjustments, especially considering the text justification. Please see standards https://journals.plos.org/plosone/s/tables

Response: We agree with you. Therefore, we have added and revised this in the Methods section. This information can be found in the Materials and Methods section (Assessment of finger-to-ground distance, back muscle strength, and pelvic tilt during gait). 

"Repetition of information in the text and description of the table. Note that there is redundancy in explaining the content of Figure 1 in the text (lines 132-134) and below the table. Please adjust the text below the table to make it more accurate and concise" and "*The formatting of Table 1 needs to be reviewed and adjusted, especially considering the text justification. Please refer to the standards at https://journals.plos.org/plosone/s/tables" have been corrected in accordance with the reviewer's comment.

Comments

In the Discussion section,

*Between lines 258-264 there is an excessive use of the term “However” which has made the text confusing. Please adjust this part of the text, seeking to express the central idea in a more objective and reader-friendly way.

*In lines 257-258 (“Hence, boys may be exposed to potentially more dangerous and strenuous sporting activities than girls.” and in lines 262-263 (“The differences in these results may be related to lifestyle and exercise habits in different countries) claims are not supported by the results of the present study. It is strongly suggested that references be added to support such claims.

Response: Thank you for your overall valuable comments concerning our study. 

*Between lines 258-264 there is an excessive use of the term “However” which has made the text confusing. Please adjust this part of the text, seeking to express the central idea in a more objective and reader-friendly way: 

Response: We apologize for this. We have had our revised manuscript reviewed by a native English editor and attached the Certificate of Language Editing and Proofreading. 

*In lines 257-258 (“Hence, boys may be exposed to potentially more dangerous and strenuous sporting activities than girls.” and in lines 262-263 (“The differences in these results may be related to lifestyle and exercise habits in different countries) claims are not supported by the results of the present study. It is strongly suggested that references be added to support such claims: 

Response: We agree that the references be added to support such claims. We have added this information and cited the relevant literature in the discussion, as suggested.

---

## [Decision Letter · Decision Letter 1]

12 Oct 2023

Relationship Between Low-Back Pain and Flexibility in Children: A Cross-Sectional Study

PONE-D-23-20765R1

Dear Dr. Ito

We’re pleased to inform you that your manuscript has been judged scientifically suitable for publication and will be formally accepted for publication once it meets all outstanding technical requirements.

Kind regards,

Mohamed El-Sayed Abdel-Wanis, Ph.D.

Academic Editor

PLOS ONE

Reviewers' comments:

Reviewer's Responses to Questions

**Comments to the Author**

1. If the authors have adequately addressed your comments raised in a previous round of review and you feel that this manuscript is now acceptable for publication, you may indicate that here to bypass the “Comments to the Author” section, enter your conflict of interest statement in the “Confidential to Editor” section, and submit your "Accept" recommendation.

Reviewer #1: All comments have been addressed

2. Is the manuscript technically sound, and do the data support the conclusions?

Reviewer #1: Yes

3. Has the statistical analysis been performed appropriately and rigorously? 

Reviewer #1: Yes

4. Have the authors made all data underlying the findings in their manuscript fully available?

Reviewer #1: Yes

5. Is the manuscript presented in an intelligible fashion and written in standard English?

Reviewer #1: Yes

6. Review Comments to the Author

Reviewer #1: Thank you for allowing me to review this manuscript. The manuscript adhere the PLOS ONE Data Policy. The aim of this study was to identify the prevalence of low-back pain and its relationship with physical function among elementary school students. This is an interesting research topic with potential utilization across disciplines and relevant to the journal.

The authors responded to and modified all requests for corrections to the manuscript.

7. PLOS authors have the option to publish the peer review history of their article (what does this mean?). If published, this will include your full peer review and any attached files.

Reviewer #1: **Yes: **Sílvia Maria Amado João

---

## [Editor Report · Acceptance letter]

3 Nov 2023

PONE-D-23-20765R1 

Relationship Between Low-Back Pain and Flexibility in Children: A Cross-Sectional Study 

Dear Dr. Ito:

I'm pleased to inform you that your manuscript has been deemed suitable for publication in PLOS ONE. Congratulations! Your manuscript is now with our production department. 

Kind regards, 

on behalf of

Prof. Dr Mohamed El-Sayed Abdel-Wanis 

Academic Editor

PLOS ONE